# Collective and Social Representations on Nature and Environment: Social Psychology Investigation in Rural Areas

**Iulian Dincă [1,](ORCID), Dragoș Dărăbăneanu [2](ORCID) and Ionuț Mihai Oprea [2]**

1   Department of Geography, Tourism and Territorial Planning, University of Oradea, Universității Street, No. 1, 410087 Oradea, Romania
2   Department of Sociology and Social Work, University of Oradea, Universității Street, No. 1, 410087 Oradea, Romania; ddarabaneanu@uoradea.ro (D.D.); ioprea@uoradea.ro (I.M.O.)
*   Correspondence: idinca@uoradea.ro or iulian_dinca@yahoo.co.uk

**Abstract:** This is a qualitative research based on a phenomenological perspective of understanding, that aim to captures the way in which the population of rural areas from the western part of Romania understands the terms of nature and environment. Starting from valuable scientific studies related to the relationship between man and nature, we propose an original interdisciplinary approach that combines social methodology with a geographical, ecological and land use perspective. This study aims to identify the forms in which social representations about nature and environment are outlined on the level of rural areas people perceptions. As Romania is a European Union member state, its rural areas have seen transformations and changes in detail that reflect in the environmental-geographical ambience typical of the three main relief types (mountains, hills and plains), the mixed geomorphological type, its residents' basic aspirations and conscious attitudinal and behavioral levels. The two study benchmarks are the notions of nature and environment, raising perception sensitivities and everyday concerns belonging to the residents of the rural areas surveyed. The administrative unit of Bihor County, belonging to the northern half of the Crișana Province and comprised of rural communities in 97 villages, was selected as the study's target area. These villages were selected in such a way that they had to meet the requirements of balance and diversity of local environmental conditions, land use and the result of changing their land cover and the socio-geodemographic conditions of the population. A series of 1576 questionnaires were administered to subjects who are over 18 years old and are aware of the reality of their places. The results of the applied tests (Levene's test) show that the concrete factors of daily activities are very good predictors of the relationship between man and nature.

**Keywords:** collective understanding; nature; pro-environment behavior; rural area; ecology

## 1. Introduction

Since the fall of communism in 1989 and Romania's integration into the European Union in 2007, the Romanian village and its inhabitants are at a crossroads in which the evolution of the nature-individual level is divided between two evolutionary-dynamic tendencies that particularize it. This is either about highlighting the antagonism between mass depopulation through migration, declining birth rates and aging (especially in the case of small hill or mountain villages and their lack of change for urban infrastructure and economic investment) and reverse migration from the city to the village (the case of the peri-urban area, located in the vicinity of culturally and economically significant cities); or the dispute between traditionalism (e.g., crafts, agricultural techniques sometimes by archaic methods, practicing religion, customs and traditions, advanced socialization of individuals) and exacerbated modernism (e.g., the assimilation of Western European models of new home architecture, new communication and transportation techniques). Among the possible negative consequences of such an evolution of many Romanian villages is the alteration, or even the irreparable loss, of the identity of the individual or community,

including their material and cultural heritage. These situations are of great interest to us, following them in Northwestern Romania, in Bihor County from Crișana Province (see Figure 1), in different representations related to nature and the environment. In such a depository territory, what matters, in the dimensioning of the relevance of nature and rural environments, is the experiences of inhabitants in nature, the collective perception of nature, the village–nature interaction, and the knowledge of the pro-environmental attitude. This kind of reference elements for the analyzed rural area refer to an ecological-geographical reality, in which they interfere and relate to other types of environments, subordinated thematically and organizationally to pure or humanized nature. It gives meaning to the portfolio of conservatives, of decoration and of human life provided by the local nature, by virtue of a real trend of transformations with different intensities, and in full agreement with the idea of representing a part of the still-traditional rural model.

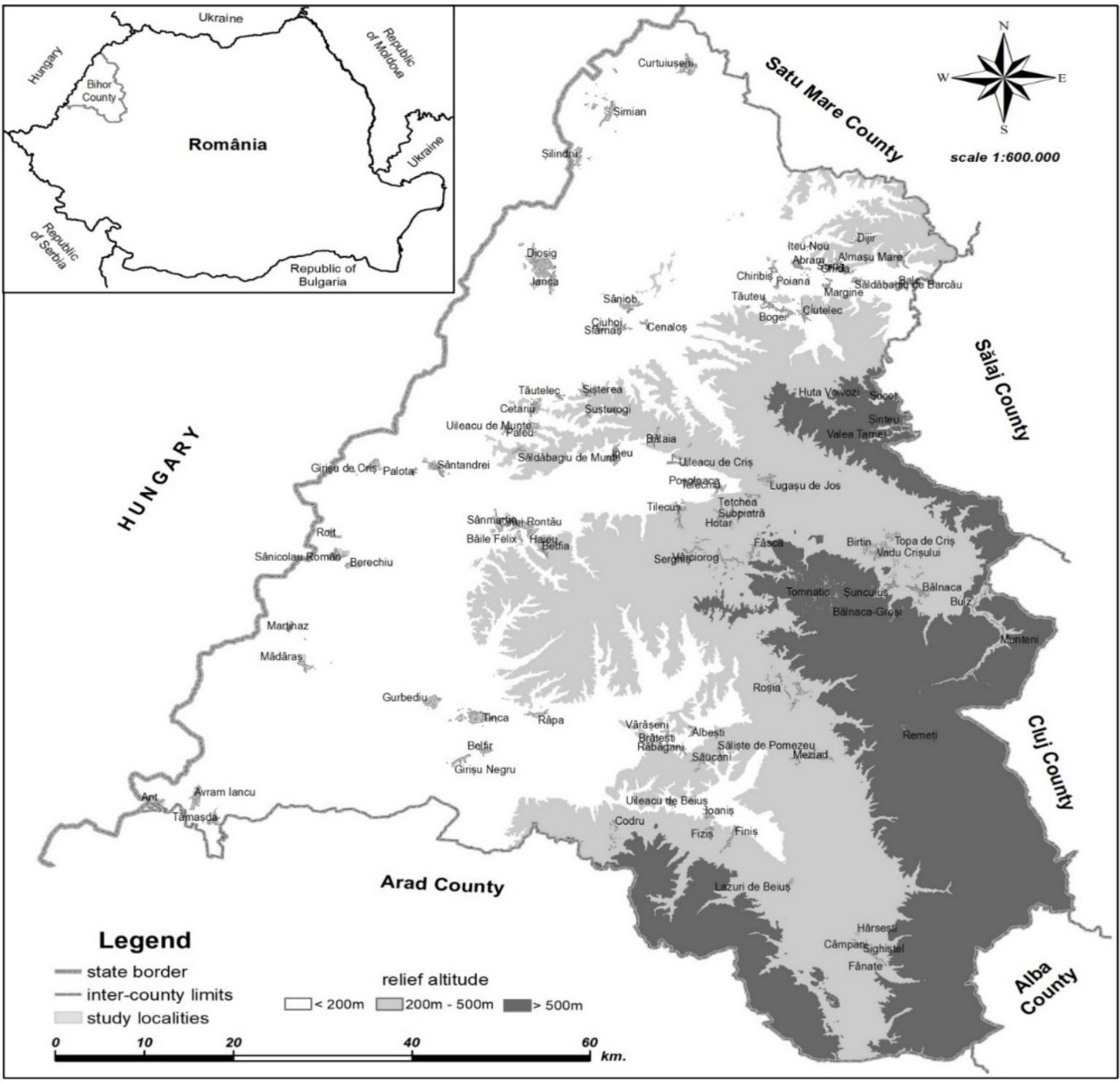

**Figure 1.** Geographical placement of villages on relief types where the surveyed subjects live. Inset, the position of the studied villages within Romania.

These aspects highlighted our interest in this geographical area, an additional reason being the fact that they did not find much echo in Romanian or foreign scientific literature. The relevant research, exploring Romanian space and particularly Transylvania, focuses less on the exhaustive, thorough treatment of nature and environment of the rural individual. With a few exceptions [1–12], the Romanian rural space is analyzed rather from the perspective of a natural capital of village landscapes [13], mineral resource exploitation lifestyle [14], of the predictions and challenges which villages face [15], of rural communities living in Natura 2000 areas in Bihor County or in southern Transylvania [16,17], of sustainable agriculture via family farming [18], of ecological measures to protect the land [9], and of the phytocenotic diversity of grazing, ploughable forest lands [19,20].

The study aims to bring the forms of nature representation and environment at individual and collective levels into basic and scientific knowledge using a psychosocial and geographic interdisciplinary approach, by studying the views of rural community members in 97 Bihor County villages. This kind of a study in rural Crișana exceeds the need of capturing the Romanian rurality, understood from a purely historical, geographical, demographic or economic perspective [13,16,21]. It is an initiative to overcome the constructive idealization of a factor or a group of factors that enhance the villages from this part of Romania. This project studies a return to the qualitative sociability between the villagers and their environment, beyond the culture of living, the more or less ecological quality of the local soils as production subsystems, of the economic exploitation in the space of the regional tradition [1] and the acceptance of entering into modernity by Transylvanian villages [22], as well as those from Crișana [16].

The studied views of rural people on nature and environment take various forms that faithfully reflect a wide range from surprising intimate physical and material environment processes in which they live, up to the exploration of immaterial, spiritual, cultural and aspirational conditions. These forms of expression result from differences and similarities between geographical areas, starting from what is both particular and common from a geo-economic and psychosocial point of view, but also from concepts perceiving villages as a social or collective construct. From showing the trend toward diversification of the requirements of the inhabitants and the transformation of the Romanian village through the power of influence of technology [23], to the transition toward sustainability of rural communities in Transylvania, including by highlighting risks, difficulties, compromises and the acceptance of opportunities [13,24], the nature and environment of the village goes through an almost effervescent and visible route. A study highlights the problems of the Romanian village related to the emigration of residents to EU countries and the marginalization of villages [25], but other studies also find solutions so that villages become animated by appropriate measures. This includes: from the tourism development of water mills [26], independent household cellars and buried in hilly mountainsides [27], to exceeding the condition of horses in the household from their purely patrimony representation to their integration into the agreement [28], the use of natural grassland of high quality value, the biodiversity of which must be preserved [19], all of which can help to practice agro tourism [29].

The practical importance of this article lies in the fact that it fills a thematic gap, offering the opportunity to apply a working model for Romanian scientists and for public administration employees as well (based on our questionnaire and results), for a more concrete, non-reductionist understanding in an ecological-scientific way, of what currently the nature and the environments of the rural inhabitants mean. By applying it to the whole country, it is possible for scientists to discern trends at the level of villagers' representations, at the level of possible changes in socializing with their nature and environments. Using data from villagers, for the administrative factor the study will count for sustainable development by knowing information about the level of education of rural inhabitants through their pro-environmental attitudes; understanding how involved they can be in environmental activities concerning their emotional affinities towards nature; and know-

ing about the appreciative activities and consumption activities relating to nature and rural environments.

It will be just as important to know the appropriate gradual-quantitative dosage, respectively, of how "natural", how "anthropic" and how much "immaterial part" must be contained in the spatial planning plans, the urbanism and strategic plans and the sectoral development plans in the case of town halls, county councils or rural development agencies.

The main objective of the research is to identify how rural people in Bihor County understand the term of nature. This concept's representation model gives sufficient precision to the interest people have in this socio-geographical area in nature, given that the attitudinal specificities lead to behavioral expressions in an area where the balance of human–nature relations, the aesthetic and ecological future of rural areas in northern Crisana explicitly depend on human action. The next stage of the analysis starts with the introduction of four variables referring to the concrete way of representing what nature means as a "place around us". The subjects were asked to specify to what extent they were interested in several aspects characterizing the main things with which they come into direct contact in everyday life. These describe the phrase "place around us" in particular terms. By performing this analysis, we aim to obtain information about the degree of involvement of respondents in relation to nature. We start from the hypothesis that if the level of nature representation is more elaborate, then the degree of individual involvement is stronger. Furthermore, a high degree of individual involvement determines the behavioral patterns of conservation and support for nature. Another objective of this study is the customization of social representations about nature and environment. For this, the concept of environment is introduced as a particular form of nature representation. The notion of environment introduces a perspective of aesthetic features in the relationship between people and nature. We start from the hypothesis that the aesthetic perspective is a motivating factor for nature conservation activities.

## 2. Understanding Geographical Area as a Case Study

The nature and the environments of these villages, understood as local and subordinate regional socio-ecological subsystems [13,30], are geographically located in northwestern Romania (see Figure 1), Bihor County (7544 km$^2$), and belong to the northern half of the Crisana historic province. These villages are spatially distributed both on the 3 large relief types (mountains, hills and plains) and on spaces belonging to the mixed type (plains-hills, hills-mountains) (see Table 1).

**Table 1.** Number of villages in the study and distribution by relief.

| Plain Villages | Plain-Hill Villages | Hill Villages | Hill-Mountain Villages | Mountain Villages |
|---|---|---|---|---|
| 55 | 6 | 26 | 3 | 7 |

This choice was based on the different meanings of the components of nature and environment in which the people of this rural space live, nature and environments in which they live and with which they (also spiritually) interact and whose resources they use. The villages benefit mostly from favorable climatic conditions, with annual thermal averages in the plains and on the hill-plain contact varying between 10–10.3 °C, and with the average precipitations between 630–680 mm/year. In the hill–mountain and mountain areas, there are slightly more restrictive conditions (more humidity-averages of 700–1100 mm/year; cooler temperatures—annual averages of 6–10 °C; slightly windy—wind with an average speed of 1–3 mps). The villages have different demographic sizes, from 200–600 inhabitants in the mountains, to 700–1100 inhabitants on the hills and up to almost 1500 inhabitants in the lowland villages, with a numerical plus for the villages that occupy a mixed morphological position. This situation is also reflected by the categories of land and natural resources, the upper steps (mountain-hill), with overwhelming proportions of coniferous and deciduous forests, hayfields and pastures (70–90% of the surfaces), which are slightly

arable. Lower forms of hill-plain relief are dominated by arable land (60–70%), followed by meadows and pastures (15–17%), orchards and gardens (5–15%). Other natural resources, which are associated with the studied countryside, are the thermal waters (in the villages from the central part of the county), the oil and lignite (in the villages from the northeastern part of the area), the still water springs (in the mountains) and the mineral water (in the plains) and the lakes (from the mountains to the plains). The villages, belonging to this study, are overwhelmingly dominated by the individual habitat, the houses being built in a contemporary style. The average household is developed on 500–600 sqm, including other outbuildings in the yard (summer kitchen, stable and hay storage, cellar, storage for equipment of small farm) and a garden for vegetables and other greens. The villages are scattered and dissipated on the mountains and hills, and are gathered and elongated mainly on the plains, but partially on the hills too. The main streets are slightly wider (6–9 m), while the secondary streets (dirt roads) are located at the edge of the settlements and they are relatively narrow (3–5 m).

## 3. Literature Review

### 3.1. On Representations in Social Psychology

The dependence of public or private institutional environments on society's perception of the actions, services and goods offered is on the rise. Durkheim [31] developed the notion of collective representations on the basis that "psychical life is a continuous course of representations which we do not know where they begin and where they end" [31].

These representations are understood by individuals through experience in relation to the social environment. This creates understanding and interpreting perspectives which get enforced at the social level and ultimately condition integration into the community. Representations become stakes and personal success paths by gaining real social motivation. This mechanism for the generalization of personal representations leads Durkheim to introduce the term 'collective representations'. Some contemporary authors [32] have distinguished between the collective representations and social representations, in the view that the first is a widespread concept that does not distinguish between different forms of building common perceptions. Political views, tradition, ideologies or scientific approaches make up, to an undetermined extent, collective representations, ambiguously conditioning the relationships between individuals and society. Class affiliations and socio-demographic features are attributes which usually influence the relationships between people and society. The Durkheimian notion, however, does not take into account these specificities and imposes itself as an inflexible form of social homogenization [32].

A comprehensive perspective of the notion of social representations was offered by Moscovici, apud [23]. He defined social representations as "dynamic ensembles, theories or sui generis collective sciences, intended for the interpretation and formation of reality" (Moscovici, apud [23] (p. 66)). We understand that such a concept is difficult to embody into a comprehensive enough definition, but from Moscovici's approach, one can notice the dynamic character of the phenomenon. Besides interpreting reality, social representations contribute to the formation of reality, which justifies the importance of this phenomenon in the dynamics of human community processes and at the same time its importance in explaining and interpreting social realities.

The context of pollution and the problems that arise as a result of this phenomenon have led scientists to be increasingly concerned about the relationship between people and nature. Thus, different strategies for stimulating pro-environment actions were developed. In this context, the Social Identity Model of Pro-Environmental Action (SIMPEA) is significant. This model of understanding social identity is specifically focused on the human–environment relationship and how this relationship contributes to strengthening social identity. According to SIMPEA, there are four basic processes that determine social identity: emotions and motivations; group identification; rules and objectives of the group; and collective effectiveness [33]. Social identity is a motivating factor that determines the crystallization of attitudinal forms and the expression of collective behavior models, which

is why the cited authors argue for the intervention of social identity as an effective way to determine pro-environment behaviors. Other studies have analyzed the relationship between the place of living attachment and pro-environment behavior. It is specified in this context that there are two forms of attachment: a civic attachment and a natural one, depending on the criteria of manifestation of the place attachment [34]. However, the specialists consider that the time spent in a locality and the perspectives that an individual has possibility to carry out activities in other parts are factors that determine the intensity of the place attachment. Moreover, throughout life, this feeling is changeable, which underlines the fact that it is not the best idea to rely on place attachment in the context of promoting pro-environment behavior.

In a context of interest for going deeper into the issues of nature and environment, social representations offer the most revealing information. This thoroughness is necessary despite signals from researchers studying the connection between nature and environmental management, of the like: "This review suggests that there is quite some overlap in the literature on CNT concepts, and that more effort needs to be made towards multidisciplinary research..." [35] (p. 1). A significant analytical tier is based on research into the concern on favorable environmental attitudes [36], nature and ecological spirit of the individual and of the group in local context, generating a social environment [20]. Another orientation of studies on nature–environment from the perspective of representative models is the empirical one. One model is rural beliefs and figurative ways of life in Greek rural areas (favorable views on eagles, negative views on foxes and wolves, acceptance of a compromise between nature and the humanized part) [37]. Sometimes, the effects of public policies and the reflection of environmental components, such as the three main components on social representations linked to the Chiampo River in northeastern Italy, are found in analytical models. These components are images of the river, emotional experiences related to them and water use practices [38]. Some other times, social representations include pro-environment farming practices [39] as well as the environmental representation in itself. This is the case with Chinese tourists who, in the form of their touristic experiences, use the description of the spectacular physical characteristics of the components in online photography [40] or by managing the risks and stressful events that can mark their personal objectives, projects and careers [41]. Environmental representation, as a variable of influence, management and control of professional choices, refers in some research to the importance of a person's territory of origin [42]. The same environmental representation is obvious from studying articles in a local American newspaper, which outline environmental changes in climate context and general challenges arising from local individual and collective actions [43]. Cultural resonance is also used as a reference in explaining higher social representations on space, nature and the environment (e.g., aesthetic or inclusive). This is the case of a protest group and a community from within a Dutch national park, engaged in a dispute with the nature conservation agency that mainly only uses references to nature wilderness [44]. A group of studies has referred to social representations through different forms of exploration. They include the visually-figurative dominant countryside ones looking into a single dimension, namely as agricultural product suppliers [25] or the visually-narrative ones using individual experiences related to the built environment, such as children in the school bus [45]. The Portuguese countryside is also socially represented by images leading to variants in physical and inhabited space (anti-idyllic views), as an idyllic space, abandoned and disadvantaged, as a place for positive transformation and socio-economic development, and as preferred for tourism activities and exploitation of natural resources [46].

### 3.2. Nature between Theory and Subject of Anthropic Transformation

Nature is frequently disputed by different wordings and meanings. Some are related to the natural and non-natural representations appearing in wording and qualifications that try to get closer to the reality on the ground. Some are seen as relatively untouched (e.g., wild areas), others are natural human-altered or non-natural environments (e.g., urban

green spaces) [47], places of aspiration and relaxation [48]. A study finds that nearly 77% of US respondents consider themselves as part of nature, but they see nature outside any traces of human intervention, a kind of nature sacralization [49]. Nature is a summary of natural or rural scenes [50] (p. 100), an association between nature and green space, which can be understood either singularly or as an integration and an interconnection. Green space thus forms an area of peaceful social interaction and national cohesion in a Lebanon destroyed by civil war [51].

Another interesting phrasing on nature and nature's place is in its psychological relationship with humans: "It seems to be possible for people to view themselves" targeting the duality of human personality. This applies when it comes to either the responsible-partisan or the opaque-oscillating variation, in the "as a part of nature, but then define nature as the non-human world" expression [49]. The unaltered, "natural" nature, is perceived by the respondents of the study initiated by Vining et al. [49] in the form of simple non-disturbing elements (trees, wild animals, lakes, grasslands, man, etc.), while "non-natural" elements make up and generate daily life (large city, store, house, car, people, rubbish, highways, etc.) Qualifying attributes are reported together with these elements (natural: beauty, serenity, calm, etc.; non-natural: pollution, altered, populated, noisy, etc.). The link with nature is reported in various forms such as: in contact with, connected to, or part of nature, as a type of human–nature relationship [52]. A new treatment scheme in the form of a 5-tier scale (Nature Relatedness—NR) was designed from the same benchmark of the nature bond of individuals. The affective, cognitive evaluation and extended personal and experimental aspects are priced: "Please respond as you really feel, rather than how you think "most people feel [53]. Another perspective is that of encouraging normal human affiliation to the nonhuman nature category, despite the nature decrease by urbanization and industrialization [54]. Nature is expressed in other studies by emotional affinities developed for it, in the form of protective attitudes, past and present experiences [55], by expressing empathy towards it. This includes elements with reference to representative forms like work satisfaction and mental health of workers achieved through plants in a closed space and sunlight [56]. A quality nature (through natural qualities), in terms of quality life for humans, is equivalent to the landscape, but genuine nature is rather found in the form of "remains" [57]. Exposure to nature and the mental well-being state, future well-being of adults can reveal a marginal relationship with nature in childhood [58]. According to Kellert [59] (cited by [59]), people's relationship with nature can have nine basic values from an emotional, intellectual and material point of view: utilitarian, naturalistic, scientific-environmental, aesthetic, symbolic, humanistic, moralistic and negativistic.

The psychosocial perspective of the interaction between nature and society introduces the concept of supradetermination [60]. The sociological vision also sees nature as "non-human natural conditions and surroundings" in the environment [61] (pp. 1–2). Nature also appears as a subject and interlocutor of our communication in third-party aspects [15], as a transactional element [49] while also opposing the reporting between the actual nature and technological nature (e.g., video clips about nature, robot animals and immersive virtual environments) [62].

The perspective of geography is also added: "nature as inescapably social" [63] (p. 4), despite nature being treated as goods that can be sold to tourists [64,65]. Exploring the meaning of the word "natural" ultimately takes into account knowledge on people's food attitudes in 5 European countries and the United States. Plants are associated here with the idea of natural much more than animals, to which one adds the absence of human intervention or artificial substances [66]. A synthetic analysis tests nature types (fatalist), perverse/tolerant (hierarchical), benign (individualistic) and ephemeral (egalitarian), based on different levels of environmental risk perception [67].

*3.3. On Environment Only*

The environment is elementary and mentally defined in reference to our proximal space, bearing living conditions: "... those external conditions or surroundings around

people" [61] (p. 1). It is an array of biotic, abiotic and people integrated into the natural or built environment. These environments are understood by affective and aesthetic reactions [50], up to where "The rural environment is an object for aesthetic consumptions..." [68]. Rural residents in relation to, and dispute with, their psychophysical environment are characterized by perception and knowledge processes, by reference to their places and resources [69], as well as by activities characterizing the farming profession [70]. Other studies highlight the complexity of people's relationships with the environment through attitudes towards it [67,69,71], through various levels of awareness, commitment, behavior and pro-environment action [72–74]. Consequences are measured by individuals' different reactions and interventions to the likelihood of environmental threats [70] or as related to a time scale [74].

### 3.4. On Nature and Environment as a Whole

A psychosocial acceptance nature and environment highlight the individual at the natural, behavioral and social levels [17], quoted by [75]. One can talk about a two-dimensional model of the attitudinal approach: nature as appreciation and environment by protection [76].

Environment and nature are understood in a non-reductive, aspirational manner, i.e., not only what is entirely pure, unaltered but rather environment with a dose of natural, with parts of naturalness, an acceptable and reasonable compromise between artificial and natural in which the human individual lives well, resulting in a natural environment [72,77] (pp. 571–572). Nature is frequently equivalent to environment under the term of natural environment, without a clear difference in attitudes of individuals [44,72,76,77]. An equivalence can be suggestively found in the phrase 'Environment as <nature>...', where the environment is indicated as an undefined geographical agent [78] (p. 198; p. 204); [58] (p. 129). Examples of natural environments are provided by participants in the study: agricultural land, alpine floor/snow zone, sports grounds. Reynard and Coratza's opinion [35] (p. 295) is similar, placing a natural environment into a spectacular association of components—relief, vegetation and scattered hamlets—belonging to the subalpine and alpine floor.

Research highlights, by the "natural environment" phrase, the idea of caring about the environment. "People see themselves" [66] while nature sees "natural environments and related wildlife" [79]. The psychological approach of nature [80] is a reference to the protectionist, securing attachment to the natural environment and its non-human contents (non-human natural environm ent). Grass plots are associated with the natural environment and open space [52], the same way as scenes involving green (hospital parks and gardens) positively influence patients' health [71].

Opinions say that another association between nature and environment comes from the cognitive-affective affiliations of people with environmental components under dichotomous cohabitation forms such as "nature and culture" [54] or "nature versus built environments" [81]. In Western Sichuan rural areas, associating Buddhist agro-pastoral communities with natural environment is supported by 3 religious dimensions: local gods and spirits in the landscape, karma and revenge of local gods, as well as Buddhist ethics [82].

Thematic literature also notes the collection of data in the form of a poll addressed to Swiss adults [75]; questionnaires on nature filled in by Chinese and Australian tourists [65]; students in psychology [67]; Australian adults [58]; and Brazilian students [83], through interviews of rural Greek inhabitants within a natural protected area. Also of importance is the open question of Dutch nature: "Why is this place important to you?" with results showing categories such as peace, tranquility, exploration and peak [84]. A computer-based questionnaire sought to raise the awareness of Dutch respondents on the value of nature in the form of car usage [18], while a hierarchical cluster analysis values the opinion of Portuguese subjects on the representations of their rural environment [46].

A phone interview on finding out what does natural mean in the minds of respondents from 5 European countries and the US [66] plus photo voice interviews highlighting the connection between the school bus and the opinions of Canadian school pupils on built environment [45] are also noted.

## 4. Methodology

Research planning (Figure 2): The research captures the way in which the population of rural areas, from the western part of Romania, understands the term nature and environment. This is a qualitative study built on the model of phenomenological research.

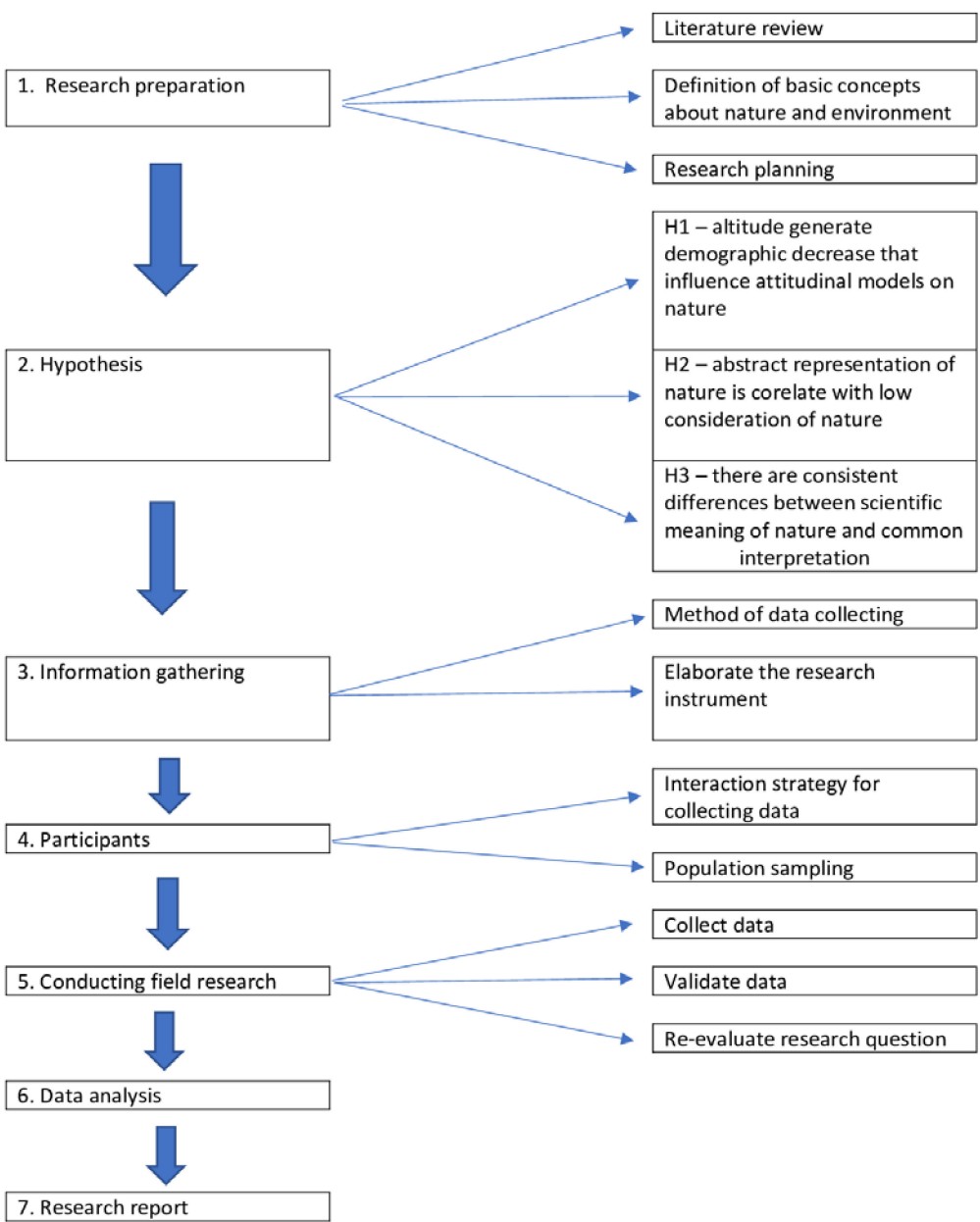

**Figure 2.** Diagram of the research stages.

The phenomenological approach studies the meaning that participants give to their daily lives, the understanding, and interpretation of everyday realities [85]. To complete, the qualitative perspective captures in-depth aspects of the ways in which research participants understand how the interaction between nature, environment, and social environment is manifested. The purpose of qualitative methods is to interpret the mean-

ings and intentions underlying human actions [86]. Thus, we aim to highlight the way in which the people of rural areas make connections between everyday experiences and natural understanding.

Hypothesis (Figure 2): As it grows the altitude, difficulties arise in the formation and preservation of villages and human communities, difficulties caused by the slightly precarious climate quality and by not particularly encouraging relief conditions, the number of villages being therefore smaller. This aspect generates a demographic phenomenon of decrease and aging of the population as the altitude increases. We expect to identify two categories of attitudinal models: one socially built out of the need to conserve nature and a second natural model, characterized by a reduced framework of intervention on nature, which is due to a simpler lifestyle. The first is manifested in low altitude communities, the second is specific to hill and mountain communities. The more abstract the social representation of nature and the environment is, the less likely it is for people to take nature into account in the activities they carry out. We started from the hypothesis that the inhabitants of rural areas have an ambiguous perspective about the environment and nature. By applying the questionnaire, we aimed to assess this, analyze the situation, and formulate conclusions related to people's relationship with nature. Another hypothesis refers to the fact that there are major differences between the scientific meaning of nature and the meaning that people offer.

Information gathering (Figure 2): For data collection, a questionnaire-based survey method was used. The research tool mainly contains closed questions that include forms of definition and understanding of the nature and environment. This method of data collection allows having a large sample due to the short time of application. On the other hand, the closed answers are due to the fact that we expected to find a certain weight in explaining the phenomena of environment and nature. In this way, we obtained in addition homogeneity of the categories of answers. The face-to-face survey allowed the operator to read the non-verbal communication and the reactions, which proved to be useful in the correct data collection.

Participants (Figure 2): The research tool includes a set of predefined terms for both nature and the environment. These definitions, proposed to subjects in the villages under examination, refer to seven nature representation aspects and six environment aspects, built in line with perception sensitivities and daily concerns, and were theoretically attributed to residents of the target area. 1576 questionnaires have been applied to the same number of subjects over the age of 18, who are closely aware of the reality of their places.

Population sampling (Figure 2): The work method involved multi-stage sampling; the study was carried out on three stages of work. In the first stage, a random sample was conducted with the municipalities belonging to both the three main relief stages (mountains, hills, and plains) and to mixed-type spaces (plains-hills, hills-mountains). The most selected villages (55 villages—Table 1) are geographically located only in plain areas and the least are located both on hills and mountains (three). This choice was dictated by numerical considerations. The second stage involved applying the random step method to select households in the field, while the last stage would see the selection of subjects who answered the questionnaires. The sampling error was approximated to $\pm 3.7\%$ with a probability of 95%.

Data analysis (Figure 2): The theoretical model of this study presents in detail the definition of the term of nature from a scientific point of view. We aimed to determine statistical indicators to understand how people define nature. By applying a Likert scale with 10 degrees, the distribution score and the uniformity of the values were obtained, in order to characterize the way nature is represented. The concept of environment is used to understand as much as possible the public perception of nature because the term 'environment' is more accurately understood by the population. In order to establish the connection between the social representation of nature and the factors that define the everyday environment, we propose two tests of significance: t-test for equality of means and Levene's test for equality of variances.

*Research Framework*

An important theoretical objective is to find studies related to social representations. Voelklein and Howarth's work [32], as well as Durkheim's perspective about the notion of collective representation [69] are theoretical aspects on the basis of which we want to understand and define social representations about nature and the environment, as well as the connections between them. Moscovici's theory [22] of social representations and the study signed by Hovardas and Stamou [37], directs us to an analysis of how climate, lifestyle or habits contribute to the crystallization of social representations related to nature and the environment. The study signed by Restall and Conrad [87] supports the idea of an interdisciplinary approach to develop a geographical, sociological and psychosocial perspective on representations about nature and the environment. The practical objective of this study is in fact to promote pro-environment behavioral and attitudinal models. From this point of view, the model of social identity construction proposed by Fritsche [33] as well as the theory of area attachment, promoted by Scannell and Gifford [34], are landmarks in the models of understanding the forms of human manifestation in its relationship with nature.

## 5. Results

In order to identify aspects of understanding nature, subjects were asked to choose the definition of the term, depending on how relevant they think it was for understanding it (Figure 3). We note that the most chosen definition of nature is "the place around us" (67.8% of all responses), which is a general perspective with a high degree of impartiality and can be seen as representative from the perspective of the representations or concerns characterizing the relationship between population and nature in the targeted social area.

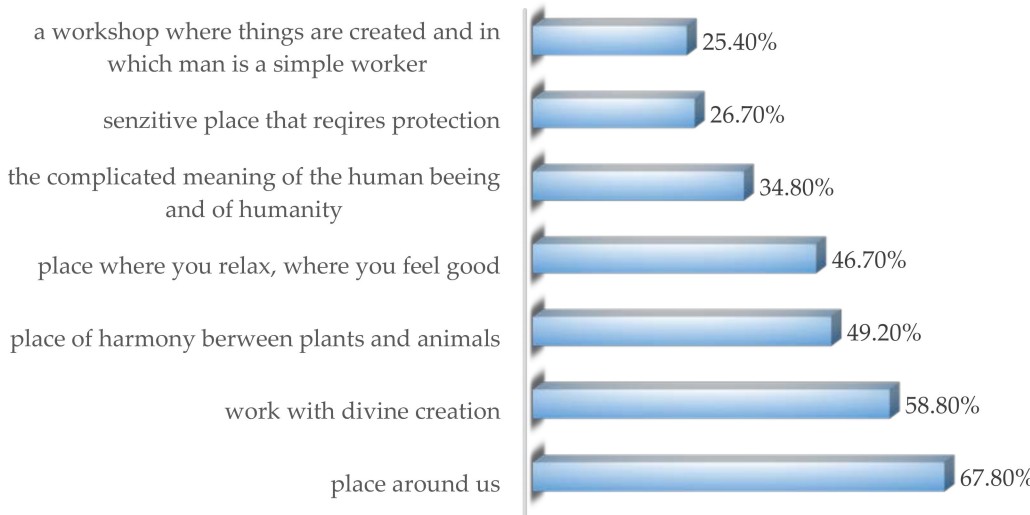

**Figure 3.** Social and collective representation of nature in the case of Bihor County residents.

Beyond recognizing the strong rooting of subjects in the pure and immediate understanding of geopolitical spatializing, these dominant responses in fact indicate attachment, knowledge and integration. The least three features of thinking and reaction are particularly found in the way subjects know how to value nature's favorable "offer" for individuals and households (Figure 3). This involves the absence or limited manifestation of hydro-geomorphological risk phenomena (rare floods and landslides) both on the main landforms and on mixed morphology (a slightly waved and hillish dominant topography), with fertile enough soil that is easy to work with in mixed crops of corn, vegetables, fodder, orchards or garden fruit trees (mainly plumtrees and apple trees) and with generous possibilities for raising sheep, goats, cows for milk, and even buffalos (Figure 4c), pigs and poultry.

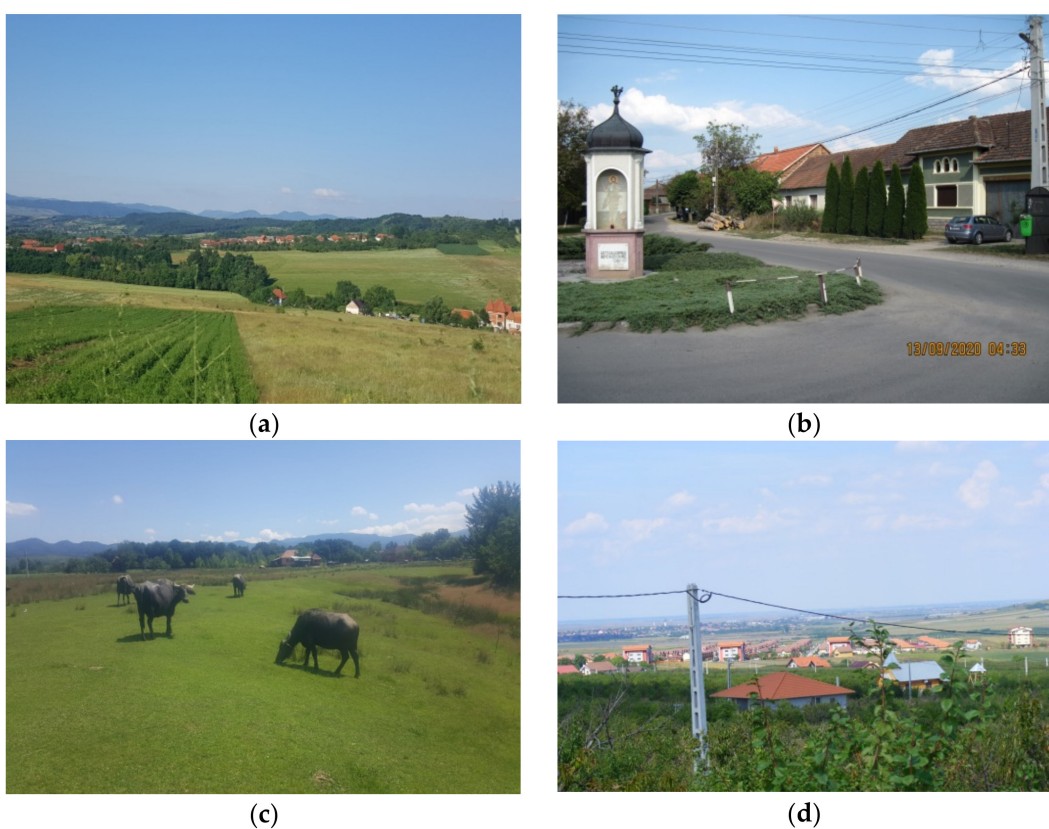

(a)                                       (b)

(c)                                       (d)

**Figure 4.** Representing nature as "a place around us" but also as a "place of harmony" between purely natural and anthropic elements: (**a**) Village of Săucani; (**b**) Village of Finiş; (**c**) Village of Meziad; (**d**) Village of Paleu.

The last plant and animal reference elements especially support residents' views (49,2% of the total—Figure 3) on the harmony of the local nature (Figure 4a) in which you can "relax, feel well" (46,7% of the answers), on top of which the anthropic element adds up (individual habitat, rarely collective habitat, roads, squares, churches, administrative buildings). Even house crowdedness and their linear street disposition (Figure 4b) or modern architecture houses and prismatic volume blocks of flats do not alter the impression of detail that the rural people have on nature (Figure 4d).

A few statistical indicators of scores obtained from a 10-degree Likert scale are presented in Table 2, measuring the subjects' representation of the quality of 4 aspects of daily life, where 1 is the farthest from the ideal and 10 is the ideal situation.

Figure 5 highlights how the inhabitants of rural environments in Bihor County represent their own idea of "environment", presented in the questionnaire in the form of "environment". The most popular meaning of the term is by far the environment defined as "the space surrounding us", chosen by 50.8% of the population (37.9% as first option and 12.9% as second choice).

This approach of those within the analyzed rural is not a random one. It underscores an evolution of individuals through school education, from where they got the right information. For some of them, contact with people with a higher vision of the world and life, people in Romanian cities or even periods of working abroad in the European Union is of importance. The two main representations of the environment as the first choice of responses (60.2%) satisfy guidelines that speak to residents about attributes, ecological and geographical features of organization and operations ranging from simple to complex. Environments pass on peace and control to the subjects. They feel they are connected to natural elements in lowlands, slightly higher fields and plains–hill areas. These offer them easy work opportunities for molisols and reddish-brown clay soils in order to obtain vegetable products, even in small lake sectors, ponds, underground water, water drainage

canals, hygrophylic vegetation or in pastures, meadows, orchards, scrubs, broad-leave tree thickets on a drier subsoil (Figure 6a,b).

**Table 2.** Statistical indicators related to the perception of nature through 4 permanent aspects of respondents' daily lives.

|  | The Most Popular Definition of Nature (Dummy) | N | Mean | Std. Deviation |
|---|---|---|---|---|
| place around us, our own household | chose the definition | 516 | 8.61 | 1.465 |
|  | did not choose the definition | 245 | 8.04 | 2.131 |
| place around us, home town | chose the definition | 516 | 6.94 | 2.113 |
|  | did not choose the definition | 245 | 6.36 | 2.319 |
| place around us, the residence area | chose the definition | 516 | 9.28 | 3.774 |
|  | did not choose the definition | 245 | 7.17 | 2.141 |
| place around us, agricultural space | chose the definition | 516 | 7.85 | 1.768 |
|  | did not choose the definition | 245 | 7.43 | 2.112 |

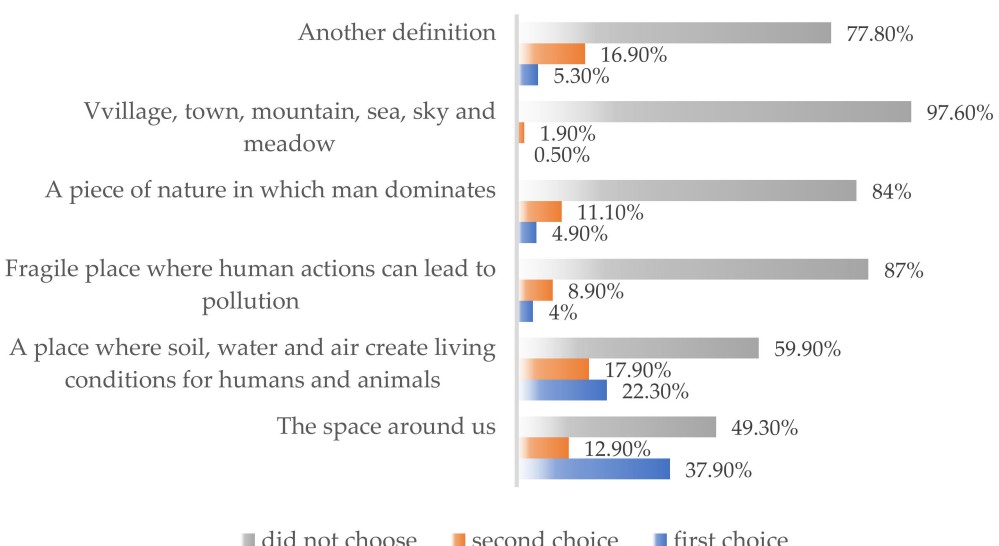

**Figure 5.** People's perception of the term of environment.

The hillside and hill-mountain stage sees environments defined by villagers' perspectives as rustic, idyllic, comfortable, and environmentally and socially secure, with grey and red brown soils having a good agricultural and forestry yield. This picture, complete with an 8–14° slope flank morphology, large enough valleys, average annual temperatures around 7–8 °C and average annual rainfall of 650–870 mm, rare extreme phenomena, is reflected in local ecological conditions which are proper for an appropriate animal and mostly human life framework. One can note the traditional prismatic family habitat (mostly houses with span or whole-hip roofs), with stables (for poultry, pigs, sheep and cows) separated from ploughable land (for vegetables, fodder, cornfields), pastures and grazing fields, roads, alleyways and fences incorporating modern but frequently natural mineral materials (sand and pebbles, lime, paling, planks, stone—Figure 6c).

Between 750 and 950 m (rarely above) one can find the hearths of the villages whose respondents appreciate their environments for life and work in ways associated with the plains, bucolic, and with a moderately wild character. They are supported by a petrography of hard or slightly softer rock configuring suspended field-like plateaus (Figure 6d), but also slope flanks above 20–30°, caves, vertical caves, and other sharp slopes (even gorges). A chilly climate, moderately humid, winters with thick enough snow layers (20–80 cm), persistent (several tens of days/year), springs, creeks cold and clear, with mills and trout farms, does not alter the well-being and place acceptance of the respondents. Other

elements of the environmental framework complete the local living environment by the usage residents give to resources such as building rocks (limestone, crystalline schists), remote trees as well as large forests within village hearths with wild fauna. Joining this picture is the village hearth with scattered houses (50–150 m from each other), with consistent grazing fields (40–60% of the village area), and flocks of sheep and cattle.

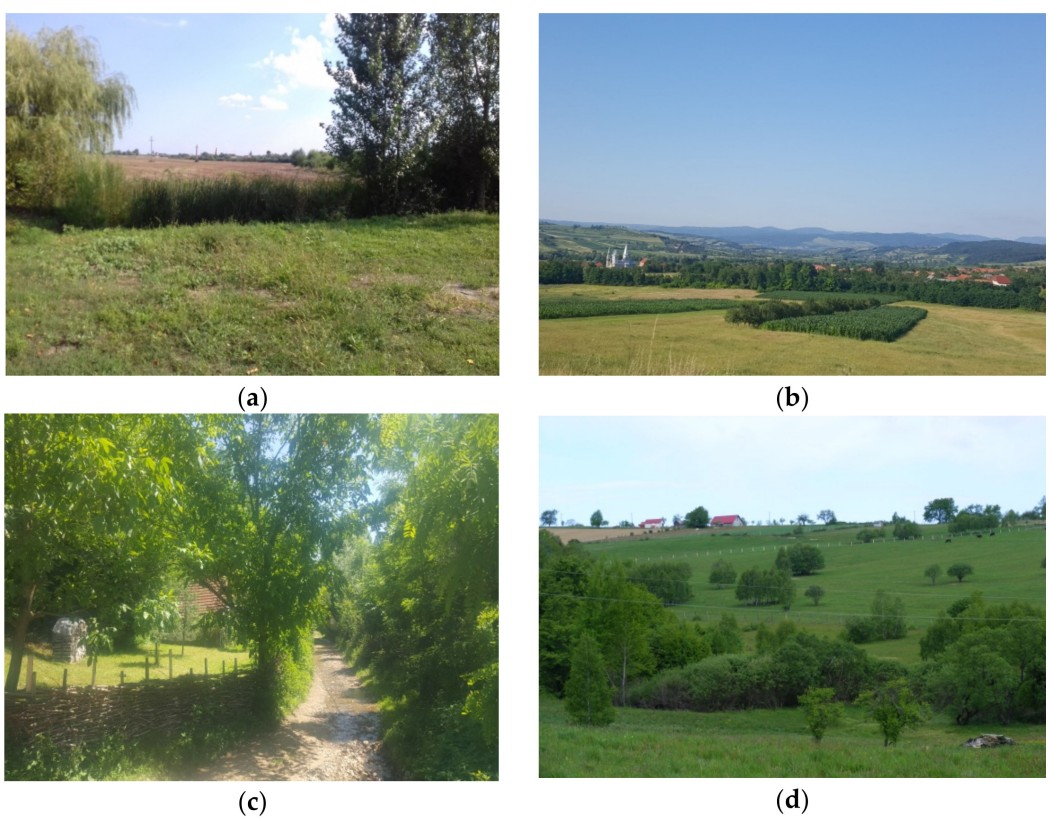

**Figure 6.** Exemplifying various types of environment in the studied area by showing the pervasive outlook and also the environment as a provider of human and animal life favorable conditions: (**a**) plains in the village of Mădăras; (**b**) mixed hills-plains in the village of Răbăgani; (**c**) mixed hills-mountains in the village of Meziad; (**d**) mountain plateau in the village of Șinteu.

We next aim to interpret the nature representations in relation to the four environment features, defined as life and development environments, characteristic of everyday life in rural environments of the Northern half of the Crisana Province, that is Bihor County. To this end, we applied the t-test to compare differences between the average of the variable defining perceptions on nature and the environments of the four variables defining perceptions on the daily environment characteristics. In other words, we aim to compare two variables: on one hand, the most widespread representation of the "nature" term and, on the other hand, we have 4 definitions of the environment with 4 different meanings, covering the term representations of more than 90% of the population.

Illustrated in Table 3 are two situations in which the perception of daily environment conditions is in a determination relationship with the perception of nature. Thus, for assessments on own households, the test of Levene (F) is statistically significant (F = 31.105 $p < 0.01$), which means $t(759) = 4.271$. Therefore, those defining nature as "a place around us" tend to develop positive representations on their own household and vice versa. At the same time, we note that the second significant determination situation is related to the perception of agricultural space. The Levene test is statistically significant (F = 13,054 $p < 0.01$, so $t(759) = 2.887$) shows that subjects from the selected sample defining nature as a "place around us" tend to positively evaluate the current state of agricultural space

in rural areas, with the category of land and their use, as an environment for individual, family and collective development.

**Table 3.** Tests on meaning of the link between the nature representation and four factors defining the daily environment term.

| Variable | | F | *p* | t |
|---|---|---|---|---|
| Environment—own household | Homogeneous dispersion | 31.105 | < 0.001 | 4.271 |
| | Heterogeneous dispersion | | | 3.756 |
| Environment—place of residence | Homogeneous dispersion | 7.097 | 0.008 | 3.430 |
| | Heterogeneous dispersion | | | 3.318 |
| Environment—area of residence | Homogeneous dispersion | 0.624 | 0.430 | 0.832 |
| | Heterogeneous dispersion | | | 1.205 |
| Environment—agricultural space | Homogeneous dispersion | 13.054 | < 0.001 | 2.887 |
| | Heterogeneous dispersion | | | 2.711 |

We thus discover the social reality in relation to a volatile, general term that is ambiguous in terms of common sense. We usually find serious differences between the scientific meaning of a phenomenon and the way it is found at the social representation level. This is also proved by the distinction, found in all sociology textbooks, between common knowledge and scientific knowledge. Our approach on social representations of the "nature" term fully confirm this. The theoretical model of our work shows nature as being a number of pure things or places, unaffected by human intervention or actions. The environment links society to nature, and defines the form of human interaction with nature. Looking at our study's data, social representations on nature in rural environments in northwest Romania do not crystallize on any scientific foundations related to nature or environment.

## 6. Discussions

Some research highlights the links between the way of people understanding the nature and the type of activity they usually perform in nature [88]. The authors talk about three types of experiences: consumption activities (in which man takes something from nature—example: fishing, hunting); mechanized activities that assume that interaction with nature is mediated by technological equipment and vehicles; and appreciative activities (hiking, studying flora or fauna, sports, etc.) Our research complements this theory by interpreting the social perspective on nature and the environment and the differences related to scientific outlook. The geographical perspective is an explanatory direction less explored in sociological research on nature, although it is a determining factor. The explanation of the difference between scientific perspective on nature and its social representation is that subjects are knowingly and immediately interfering with their nature by living and exploiting resources, seen as components of nature, from their own backyard to their land in the village. They understand the geography of nature not in a reductionist way, but in a joint and skilled manner, by the simple organization of the nature of their places through purely natural villages' proximity, which are physically dominant (80–90% of the total surface area): hillsides and mountainsides with slopes rarely over 20°, mature and thick hornbeam, oak and beech forests, tilled lands, geometric-looking pastures, livestock and easy-to-see wild animals, with which they resonate observational and aesthetically. Understanding of social identity is considered to be an effective way to determine pro-environment behaviors [33]. Moreover, the place of living is an important part of social identity and scientific knowledge highlights the relationship between the place of living attachment and pro-environment behavior [34]. Socializing with their nature, the subjects retain the living nature of physical and anthropic eco-geographical bodies, both agreeable and tonic, refreshing combinations supporting on image elements through modest to large viewing axes (2–5 km), wide angles (90–110°), elongated volumes, 2–4 main landscape

plans and an austere chromatic palette in resting countryside tones. What is also interesting is the positioning of subjects against the architecture of their surrounding nature, which they perceive as a "work of divine creation" (58.8% of total answers). A good proportion of rural residents cannot separate themselves from divinity in their relationship with nature, despite the modern times and the modernization of villages. Their vision is a creationist one but they understand their relationship with nature through the idea of divinity, including built elements such as churches and small religious buildings not destined for worship but rather perceived of as a gratitude to divinity. These include the triptychs (Figure 4b) as geographical landmarks, in certain parts of villages, of a rural nation with inhabitants marked by religion and an attachment to divinity.

We mention once again that the rural population of the northwestern part of Transylvania represents a different social environment compared to developed areas (urban or rural) from Europe. In developed areas there is a decrease in direct contacts between man and nature [70,79]. To compensate for this situation, an attempt was made to promote educational activities in early childhood, based on interactions with nature [83]. In the case of people from northwestern Transylvania, the connection with the natural environment must not be stimulated, but it is an integral part of daily life. However, we consider as a point of reference the way of defining pro-environmental attitudes as a "collection of beliefs, affect, and behavioral intentions of a person holds regarding environmentally related activities or issues" [89] (p. 458). The social and collective representation of environment specific to the population studied is oriented towards a general dimension. People's environment representation need is an extremely broad one, most people tending to include as many aspects as possible in this dimension (Figure 5). As noted, the most specific representation is the one which presents the environment as a "village, city, mountain, sea, sky and pasture". This representation is avoided by respondents precisely because of the need for generalization, thus avoiding a repetition of the components' representation they have about nature. The particular aspects of the environment are considered insufficient to describe this complex element. We see this as evidence that the notion of environment is hard to understand from a global point of view. People can explain themselves using various parts of the environment, particular cases encountered in specific contexts, but the global perspective is perceived as complex and, because of this, a general definition for it is preferred. On the other hand, we note that respondents (5.3% of the total responses) prefer a definition other than those that were proposed in the questionnaire. Yet, having been asked to formulate an alternative definition, respondents were unable to do so. This is another argument that supports the ambiguous outlook people have about the term "environment". The ambiguity of social representation has repercussions at the levels of behavioral and attitudinal patterns. The greater the ambiguity of social representation, the less likely people will consider environment in their actions. We therefore demonstrate the need to implement environmental information programs. These are expected to customize the social representations of the phenomenon and thereby raise public awareness about the environment. Yet very encouraging news in other responses from an geographical perspective and rural mentality evolution come from the group of respondents showing a correct approach and care for the decor and life framework that environments have in configuring their places (the option "place where soil, water and air create living conditions for humans and animals"—22.3%). This correct response is linked to the conceptually covering representation of the most part of the environment definition.

Studies have shown that, in addition to place of residence (urban vs. rural), there are other social factors that influence the perception of nature, such as fi age, gender, education or even political ideology. Younger people, women, and liberals tend to be more open to improve their environmental attitudes than older people, men, and conservatives. We consider, along with previous researchers, that a more frequent exposure to nature (both for adults and children) can have positive effects on pro-environmentalism. However, with regard to rural-type societies like those in northwestern Transylvania, the most efficient

way to promote pro-environment attitudes is to introduce educational programs to regulate the relationship between man and nature.

*Discussion about the Generalizability of the Findings*

An important aspect related to this study is the way in which the research and the results obtained can be significantly used in other human communities. First of all, it should be noted that a generalization of the results must take into account the degree of "urbanization" of rural areas. At the European level, rural areas in Eastern Europe are less socially regulated than rural areas in Western Europe. This is significant because the more regulated a community is, the less likely it is that the relationship between the citizen and nature will cause damage, such as pollution, destruction of ecosystems, deforestation, and uncontrolled exploitation of resources. In urban areas there are forms of monitoring and control of human activities that can influence the environment. Compliance with the principles of nature conservation is mandatory, otherwise sanctions will be applied. Therefore, the generalization of the results of this study works primarily at the level of communities with a degree of "urbanization" similar to rural areas from Romania.

What we consider applicable in general is that poorly regulated social areas in the matter of human–nature relationships have as an alternative the education control of citizens and their conviction to adopt a correct attitudinal and behavioral model. Our study offers a way to monitor the relationship between people and nature by identifying forms of social representation of nature. These forms of representation are the necessary landmarks for the strategy development in educating populations to adopt attitudinal and behavioral models that will ensure nature conservation.

## 7. Conclusions

The main objective of the study is to explain and analyze the ways in which north-western Crisana rural inhabitants relate to the notion of nature. The significance of this social representation has at least two important connotations. We think here about the ecological perspective by the influence people have, through their actions, on nature, but also from the European perspective, viewing the human relationship with nature as a reference factor in specific regulations for Romania's integration into the overall structure of the European Union.

An important point in the context of human relations with nature, in the case of rural social environments in Bihor County, is related to ownership and responsibility. The greater the citizens' undertaking, the more nature–protection civic activity measures will be better defined, more present and effective, with a view to preserving nature and fighting polluting factors. From this point of view, we see the social representation of nature as "a place around us" as a positive factor, because this nature definition formula involves a cohabitation and an integration relationship in which people acknowledge their potential for influence on the defined phenomenon. If we, on the contrary, define nature as a "work of divine creation", we suggest a fatalistic perception of the phenomenon, which is a given, which cannot be intervened upon.

Significant associations of the most encountered nature definition form and household, respectively, and the agricultural space, support the theory that north-western Crisana rural inhabitants are susceptible to personally acknowledge a relationship with the state and evolution of nature. The individual's control over his own household organization form is clearly greater than in the case of their control over their village or residential area. In terms of what concerns the agricultural space, it is also known that agricultural activities in rural Romania are widespread to almost the entire population. Inhabitants not involved in agriculture, at least at the level of fulfilling their own needs, are hardly found in traditional rural areas. On the other hand, social reality presents modern rural spaces, which are mostly to be found in the large suburbs of the cities (so-called metropolitan areas), where households which emerged in recent years are built on narrower spaces and the lifestyle of "new citizens" rarely manifest agricultural or animal raising practices. The

trend in these areas is to give up household agricultural activities and use their limited land as an area of more or less complex landscaping.

The two main representations, by weight of responses, of the environment term follow a route from general to simple, from a geographical ensemble vision to a more geographic, ecological and social perspective in detail. In most of the rural areas under consideration in Bihor County, however, the agricultural space environmental factor and its land use categories is a personalized and controllable aspect of everyday activities.

Significant relationships suggested by the statistical processing carried out, where there is a strong determination between social and collective representation of nature and the perception of agricultural and household activities, demonstrate the northern Crişana population's predilection to assume liability in relation to nature. In fact, these attitudinal patterns reflecting social representation types play a significant role in civic spirit raising and collective action strategy elaboration to be achieved in social life areas where institutional, public and private activity is not effective enough and can only exercise partial control. The relationship between people and nature is such a situation and the results provided by our study clearly show that civic education or collective action stimulating strategies, aimed at determining an attitudinal and behavioral model consistent with the idea of nature protection, are building on an advanced background in northwestern Crişana rural environments. The tendency of ownership and responsibility manifested by the majority of the surveyed population in relation to the man–nature relationship is a positive prerequisite and provides a starting point in improving natural framework conditions.

**Author Contributions:** Conceptualization, I.D. and D.D.; Methodology, I.D. and D.D.; Formal Analysis, I.D., D.D. and I.M.O.; Investigation, I.D. and D.D.; Resources, I.D., D.D. and I.M.O.; Data Curation, I.D., D.D. and I.M.O.; Writing—Original Draft Preparation, I.D., D.D. and I.M.O.; Writing—Review and Editing, I.D., D.D. and I.M.O.; All authors have read and agreed to the published version of the manuscript. All authors equally contributed to the creation and copy editing of this manuscript.

**Funding:** This research received no external funding.

**Institutional Review Board Statement:** Not applicable.

**Informed Consent Statement:** Not applicable.

**Data Availability Statement:** Not applicable.

**Acknowledgments:** We are grateful to the academic editor and the two reviewers for their competent advice and recommendations. Our thanks also go to our colleague Carlton J. Fitzgerald at New England College (USA) for his efforts in checking, proofreading, and adapting the text of the article to the rigors of the English language. We thank the University of Oradea Sociology and Geography students who took part in the field research and data processing within this study.

**Conflicts of Interest:** The authors declare no conflict of interest.

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
