# Peer review of "Collective and Social Representations on Nature and Environment: Social Psychology Investigation in Rural Areas"

_land, doi:10.3390/land10121385_

Round 1
Reviewer 1 Report
This article presents a quantitative take on investigating social depictions of nature and the environment adopted by the people in Northwestern Romanian rural areas. There is much to like about this manuscript, considering how committed the authors are to preparing this first draft. However, some missing elements of the delivery of the research require significant revisions/additions. The following comments intend to guide the authors to ensure the high-quality delivery of their otherwise rigorously conducted study.
- Introduction
- While the Introduction section has included Research Aim, there is no build-up Background story that leads to the necessity of this research (referring to the literature, why this research should be conducted? ⇒ Research Gap). The Research Aim should be based on the Research Gap to ensure the original contribution of this article.
- Besides, this section requires a couple of Research Questions that would be addressed by this study in the pursuit of filling the Research Gap and achieving the Research Aim. Research Questions would be the points of departure from the background story (Introduction) to the core content of this research. As an example of the making of Research Questions for social science research, the authors may briefly check this article ⇒ DOI 10.1177/1350508410372151
- Literature Review
- The current Sub-sections 1.1-1.4 are more appropriate to be posited as an independent Literature Review section. Please move and group them into a new section (Literature Review).
- The Literature Review section should end with a sub-section proposing the Research Framework. The research framework should indicate the presumed interconnection between included theories/concepts/variables. This sub-section is essential to wrap up the literature review and show how the authors intend to use reviewed concepts/theories/variables throughout this research.
- Materials and Methods
- This section still lacks the systematic explanations of Research Design. While there are mentions about research activities, the systematic design of the research as a whole has not been provided. The authors should add a new sub-section (Research Design) to explain the step-by-step of this research. For each stage, the authors should explain its Objective(s), Method(s) being used, and its Expected Outcome(s). This sub-section is extremely important as proof for readers that this research was conducted systematically and has included all necessary activities and methods.
- Discussion
- I am a bit surprised that this social science research has not provided an independent Discussion section. Basically, this section should compare and contrast each key finding of this study to the results of relevant published studies. Considering the rapid progression of social science studies (the applicability of novel knowledge is about 5-15 years), this Discussion section should be the de facto proof of the original contribution of each key finding from this study to relevant body of knowledge.
- Since the Discussion section contains a thorough compare and contrast process, this section is typically longer than the Conclusion section.
- Figures
- Figure 4. This figure looks distorted. Please use the original ratio and enlarge the texts to deliver better readability. Besides, I suggest the authors separate the bottom table into an independent Table for better information delivery.
- All figures. Please remove the textured backgrounds to deliver clearer presentations.
- Tables
- All tables. For cells containing numbers, please use the right-aligned style to make the numbers comparable between cells within the same column.
- All tables. In contrast, please consider using the left-aligned style for cells containing text for better readability.
Author Response
Please see the attachement.

Reviewer 2 Report
I have now had the chance to read and review the manuscript "Collective and Social Representations on Nature and Environment: Social Psychology Investigation in Rural Areas", submitted to Land.
The paper provides several interesting and original contributions. The focus on a specific Romanian region is valuable and provides insights into nature representations of often overlooked regions in Europe. At the same time, however, I would like to see some discussion about the generalizability of the findings: How can these findings be used to understand people's nature representations in other regions? What is unique in this area, and what could be transferred to other places?
A second issue that bothered me is that the title suggests "collective representations", but how are these investigated and refered to, theoretically? When I think of collective representations, I think about work in the social identity and place attachment tradition that could help the authors develop their argument. For example, the comprehensive SIMPEA model by Fritsche et al. (2018, Psychological Review) could provide a sound basis here. Similarly, work on place attachment, such as by Scannell & Gifford (2010, Tripartite model, Journal of EnvPsy) or on what happens when we loose things that make the place that surrounds us could provide further insights about why (collective) representations of nature and "the environment surrounding us" are so important.
A third issue concerns the statistical analysis. It is largely descriptive, which is ok, but I believe that you could provide some testable hypotheses that could then be tested inferentially.
Also, there are some formal and minor spelling errors throughout. For example, in table 3: The "p-value" can NEVER be ".000", please write "<.001" or so.
The figures a nice to look at, but do not comply to any scientific guidelines I know of, but this is to the editors and journal to decide.
Finally, although I like the narrative style of the paper, I believe it could easily be reduced by around 1000 words by looking at every sentence whether it is really necessary to make your point.
To sum up, a nice and informative paper that could be strengthened by some more theoretical rigor and more scientific conciseness.
And: I really liked the beginning of the discussion where you take the "European" perspective - maybe this could be intertwined with what I suggest in the first point (generalizability).
Author Response
Please see the attachement.

Round 2
Reviewer 1 Report
First, let me declare common words/phrases used in this review. I thought that the authors understood these common words/phrases. However, the responses they make for the first review round do clearly show that they might have different understandings of these common words/phrases.
- Section ⇒ part of the manuscript numbered X (e.g., Section 3 ⇒ Literature Review). One section should serve one purpose (e.g. Section 1 ⇒ Introduction ⇒ Only to introduce the background story, research gap, research aim, and research questions).
- Subsection ⇒ a sub of a section. Subsection is part of the manuscript numbered X.X (e.g., Subsection 3.1 ⇒ On representations in social psychology).
- Literature Review ⇒ Dedicated section of the manuscript that contains a thorough review of relevant literature to develop research framework, to choose observed variables, and to build other model-related constructs. The Literature Review section is not part of the Introduction section.
- Discussion ⇒ Dedicated section of the manuscript to argue the original contributions of a research to the body of knowledge. It means it contains direct comparisons of the key findings of a research to the results of published literature. Compare-and-contrast is the key!
- Research Stages ⇒ The steps taken by researchers in conducting a research. Therefore, "for each stage" means for one stage, not for the whole research.
Furthermore, I see that the authors have put substantive revisions for several parts of the comments. I appreciate all efforts the authors have made to address the parts. After a thorough check over the revised manuscript, the following concerns arise. In general, there are critical points of concern that are completely neglected and sometimes misleading, which, unfortunately, do not help to improve my valuation over this manuscript.
- Section 1: Introduction
- This section begins by letting readers know what they intend to do (first sentence). It raises more questions rather than information. There is no build-up story of why the "elementary need" arise at the first place? Does it mean that the authors just want to know something but do not understand why they need to do that? What was the logical and linear story behind the need? The "elementary need" should come after everything is explained. It is the product of a thorough scientific thought process that leads to the necessity of this research.
- The second sentence suddenly raises "questions". While they refer to the Sandberg & Alvesson's article (2010), I am afraid the authors do not get the essence of the article. The way the authors provide research questions does not have anything to do with the article they cite. The questions come out from nowhere without any scientific thought process leading to the gap-spotting that eventually produces research questions. There is no "simple" question. This is a scientific article, thus the questions must be scientifically built from a well-defined scientific thought process.
- Linearity and clarity are the keys!
- Section 2: Case Study
- Since this research is heavily case-based research, simply make the current Section 1.1 as Section 2 (Case Study).
- Please make sure that all basic information regarding the case (before the research was conducted) are provided here.
- All other information that arise from this research should not be here, because this Section emerged before the research was conducted.
- Section 3: Literature Review
- I am surprised that the authors do not understand the meaning of "an independent Literature Review section". Introduction and Literature Review serve different purposes, thus they must be separated.
- This section must include the considerations why each of the observed variables (Lines 433-435 "four permanent aspects") are necessary to address the Research Questions.
- Section 4: Methodology
- The "Research Objectives" is part of Introduction, not Methodology. In the Methodology section, the objectives, methods, and expected outcomes are for each research stage, like what was referred to in the first review round ⇒ "For each stage, the authors should explain its Objective(s), Method(s) being used, and its Expected Outcome(s)".
- The authors must explain the stages in the way they designed the research in the first place. That's why we call it "Research Design". For a clearer presentation, please consider providing a Figure of the research stages, and explain each part of the figure in the text.
- Since this is a Methodology section that emerged before the study was conducted, the authors should not include any primary data/result.
- I am also surprised that the authors do not cite any methodological articles to build their methodology. How did the authors decide to use method A but not B? Why A+C but not A+D? Does it mean that the authors brilliantly developed every single method used in this research???
- Section 5: Results
- Results presented in this section must be correlated and ordered according to Research Objectives and Research Questions from Section 1. It would deliver a logical explanation of the results that is fitted to the linearity of this research.
- Section 6: Discussion
- While the addition of a new Discussion section is appreciated, the content does not have anything to do with the place of the key findings among the findings of published literature. I am surprised that there is no single citation in this section, meaning that this section now contains mere talks about the results of this research. There is no proof that this research contributes anything new to the scientific literature. In that sense, why bother accepting/publishing this article?
- From the first review round ⇒ "this section should compare and contrast each key finding of this study to the results of relevant published studies ..... original contribution ..... to relevant body of knowledge."
- Please do a thorough and comprehensive Discussion!
- Section 7: Conclusions
- Please simplify this section without reducing the richness. The authors can simply consider using three paragraphs:
- Paragraph 1: Summary and results of this research
- Paragraph 2: Key findings that answer research questions (match certain findings to their specific research question).
- Paragraph 3: Managerial/policy implications that arise from each key finding.
- Please simplify this section without reducing the richness. The authors can simply consider using three paragraphs:
Besides the substantial concerns on the content, there are additional concerns:
- Language.
- I doubt that the authors have asked a native English speaker to check the language used in this manuscript since elementary mistakes like commas for decimals are still around. Please do ask a professional English editing service to do a thorough language checking.
- Tables.
- Tables 2 & 3. Please remove unnecessary columns.
- Figures.
- Please add one new figure for Research Design.
- Other cosmetics.
- Please use dots (".") instead of commas (",") to separate decimals!
- What does the author intend to deliver from the dashed lines at Line 331?
- Supplementary file.
- Why did the authors attach a CV as the Supplementary File for this manuscript? There is nothing valuable to see that relates to this research.
Reviewer 2 Report
After reading the revision, I found that the authors were responsive to my comments and suggestions.
However, I do really recommend to submit the manuscript to a professional language editing service. Not being a native speaker myself, I know that it is often hard to prepare a flawless piece of work, language-wise, and have often benefited form native editors thorughly reading it.
